# A Flexible Regression Modeling Approach Applied to Observational Laboratory Virological Data Suggests That SARS-CoV-2 Load in Upper Respiratory Tract Samples Changes with COVID-19 Epidemiology

**DOI:** 10.3390/v15101988

**Published:** 2023-09-23

**Authors:** Laura Pellegrinelli, Ester Luconi, Giuseppe Marano, Cristina Galli, Serena Delbue, Laura Bubba, Sandro Binda, Silvana Castaldi, Elia Biganzoli, Elena Pariani, Patrizia Boracchi

**Affiliations:** 1Department of Biomedical Sciences for Health, University of Milan, 20133 Milan, Italy; laura.pellegrinelli@unimi.it (L.P.);; 2Department of Biomedical and Clinical Sciences (DIBIC), University of Milan, 20133 Milan, Italy; 3Department of Biomedical, Surgical and Dental Sciences, University of Milan, 20133 Milan, Italy; 4Fondazione IRCCS Ca’ Granda Ospedale Maggiore Policlinico, 20122 Milan, Italy; 5Data Science and Research Center (DSRC), L. Sacco, “Luigi Sacco” University Hospital, University of Milan, 20133 Milan, Italy

**Keywords:** regression modeling approach, SARS-CoV-2 load, upper respiratory tract, vaccination, quantile regression, regression modeling approach, Kaplan–Meier curve

## Abstract

(1) Background. Exploring the evolution of SARS-CoV-2 load and clearance from the upper respiratory tract samples is important to improving COVID-19 control. Data were collected retrospectively from a laboratory dataset on SARS-CoV-2 load quantified in leftover nasal pharyngeal swabs (NPSs) collected from symptomatic/asymptomatic individuals who tested positive to SARS-CoV-2 RNA detection in the framework of testing activities for diagnostic/screening purpose during the 2020 and 2021 winter epidemic waves. (2) Methods. A Statistical approach (quantile regression and survival models for interval-censored data), novel for this kind of data, was applied. We included in the analysis SARS-CoV-2-positive adults >18 years old for whom at least two serial NPSs were collected. A total of 262 SARS-CoV-2-positive individuals and 784 NPSs were included: 193 (593 NPSs) during the 2020 winter wave (before COVID-19 vaccine introduction) and 69 (191 NPSs) during the 2021 winter wave (all COVID-19 vaccinated). We estimated the trend of the median value, as well as the 25th and 75th centiles of the viral load, from the index episode (i.e., first SARS-CoV-2-positive test) until the sixth week (2020 wave) and the third week (2021 wave). Interval censoring methods were used to evaluate the time to SARS-CoV-2 clearance (defined as Ct < 35). (3) Results. At the index episode, the median value of viral load in the 2021 winter wave was 6.25 log copies/mL (95% CI: 5.50–6.70), and the median value in the 2020 winter wave was 5.42 log copies/mL (95% CI: 4.95–5.90). In contrast, 14 days after the index episode, the median value of viral load was 3.40 log copies/mL (95% CI: 3.26–3.54) for individuals during the 2020 winter wave and 2.93 Log copies/mL (95% CI: 2.80–3.19) for those of the 2021 winter wave. A significant difference in viral load shapes was observed among age classes (*p* = 0.0302) and between symptomatic and asymptomatic participants (*p* = 0.0187) for the first wave only; the median viral load value is higher at the day of episode index for the youngest (18–39 years) as compared to the older (40–64 years and >64 years) individuals. In the 2021 epidemic, the estimated proportion of individuals who can be considered infectious (Ct < 35) was approximately half that of the 2020 wave. (4) Conclusions. In case of the emergence of new SARS-CoV-2 variants, the application of these statistical methods to the analysis of virological laboratory data may provide evidence with which to inform and promptly support public health decision-makers in the modification of COVID-19 control measures.

## 1. Introduction

Three years after the declaration of the coronavirus disease 2019 (COVID-19) pandemic by the World Health Organization (WHO) on 11 March 2020, over 685 million people have been infected by SARS-CoV-2 and more than 6.8 million people have died [1].

SARS-CoV-2 infection can be asymptomatic or can range from a mild to a life-threatening and fatal disease; clinical progression may depend both on the host individual characteristics, such as age, the presence of comorbidities or underlying medical conditions, and vaccination status [2,3,4], and on viral characteristics (i.e., virulence, ability to evade immune response) [5].

Understanding the duration of virus shedding is critical to controlling SARS-CoV-2 spread and countering the pandemic [6].The viral load progression in upper respiratory tract samples may be an indicator of infectivity [7,8];; viral load peaks within a week from symptoms onset and then follows a relatively consistent downward trajectory, with viral load levels slowly decreasing over the following one-to-three weeks to achieve negativization (i.e., no SARS-CoV-2 RNA detected by real-time reverse polymerase chain reaction, rRT-PCR) [9]. As a respiratory virus, the SARS-CoV-2 shedding dynamics in the upper respiratory tract may provide insights into the epidemiological characteristics of the infection [10]. Moreover, several studies have associated high viral loads with a more severe disease outcome [11]. By contrast, other studies have reported no differences in viral loads by age, presence of symptoms, SARS-CoV-2 variant, and vaccination status [12,13,14,15]. The hypothesis that individuals with mild and severe COVID-19 outcomes may show higher viral loads than asymptomatic ones, also reflecting a different length in the time of negativization, has been extensively discussed without a clear conclusion.

Information on SARS-CoV-2 load at different time points of the infection—including in asymptomatic individuals—will support the clinical interpretation of rRT-PCR test results. Some studies have proposed that a rRT-PCR cycle threshold (Ct) value of 35 can discriminate whether a person is still infective [9,16].

In Italy, the COVID-19 vaccination campaign started on 27 December 2020, to reach a coverage of nearly 70% in December 2021. In December 2021, the SARS-CoV-2 variant of concern, B.1.1.529 (“Omicron variant” as labeled by the WHO), became predominant [17]. This variant has a shorter incubation period and a higher transmission rate than previous variants [18]. In populations with high COVID-19 vaccination coverage approaching a return to normality, and in consideration of the predominant circulating variant of SARS-CoV-2, it is essential to evaluate whether viral shedding duration may have different trajectories in unvaccinated and vaccinated individuals in order to apply appropriate control measures. This evaluation appears even more relevant considering that in population with access to vaccination, vaccines should reduce the SARS-CoV-2 transmission by affecting the viral load titre and viral clearance in upper respiratory samples, as reported by several studies [13,19,20,21,22]. However, the effects of vaccines could be confounded by the circulating predominant SARS-CoV-2 variant [23,24]. Therefore, understanding the timing of SARS-CoV-2 clearance in the population’s upper respiratory tract samples is pivotal to informing public health stakeholders of effective mitigation strategies, such as defining isolation and quarantine time and supporting the implementation of restrictive measures.

According to this rationale, the aims of the study were as follows:To describe, for the 2020 and 2021 winter waves, the dynamics of SARS-CoV-2 load in upper respiratory tract samples in adult subjects with positive status between the first positive test result (defined as index episode) and the last positive one.To evaluate the putative relationships between the above dynamics and demographical and clinical characteristics, i.e., gender, age, and presence of respiratory symptoms.To investigate the time of SARS-CoV-2 clearance, defined as rRT-PCR results with Ct value >35 [25], following the index episode.

To such an end, retrospective data from two cohorts of 193 and 69 adults (2020 wave and 2021 wave, respectively) whose infective status has been monitored over time were used. The two periods were considered separately because the characteristics of the infected subjects were different; in 2020, the vaccines were not available, and there was a different SARS-CoV-2 variant. Therefore, a direct comparison is not appropriate. Results obtained by applying methods of longitudinal data analysis (aims 1 and 2) and survival analysis (aim 3) provided information about two distinct endpoints: the former allows for the estimate of plausible values of SARS-CoV-2 load for subjects in the target population who are still positive after a determined number of days after the index episode; the latter one provides estimates of the probability of subjects to reach clearance in days following the index episode, thus indicating the incidence of such an “outcome” in the target population.

## 2. Materials and Methods

### 2.1. Study Samples and Individuals

We conducted a retrospective study on leftover nasal pharyngeal swabs (NPSs) collected from individuals who tested positive for SARS-CoV-2 RNA detection via rRT-PCR in the framework of the SARS-CoV-2 testing activities carried out at the regional reference laboratory for COVID-19 (Department of Biomedical Sciences for Health, University of Milan) in Lombardy, a region of Northern Italy with nearly 10 million inhabitants. Molecular testing was conducted for diagnostic purposes in symptomatic individuals (i.e., patients with respiratory symptoms who sought medical advice or were hospitalized with COVID-19) and for case-finding/screening purposes in asymptomatic individuals (i.e., healthcare personnel and nursing home residents tested every two weeks). Here, we considered two epidemic periods: the first spanning from 14 September 2020, to 11 February 2021 (2020 winter epidemic wave); and the second from 12 December 2021, to 24 January 2022 (2021 winter epidemic wave). There were differences in SARS-CoV-2 testing options during the two study periods: (i) in 2020, only molecular tests were used in diagnostic and surveillance settings, whereas antigenic tests were used as an alternative in 2021; (ii) testing for ending isolation was carried out mainly with antigenic test in 2021.

We included in the analysis only SARS-CoV-2-positive adults (>18 years old) for whom at least 2 serial NPSs were collected for clinical/epidemiological purposes. For the 2021 winter epidemic, only individuals who completed COVID-19 vaccination course were included (we excluded individuals with unknown vaccination status). The date of the first SARS-CoV-2-positive test was defined as the index episode.

The study was conducted according to the guidelines of the Declaration of Helsinki of 1975 as revised in 2008, and it was performed according to the Institutional Review Board guidelines concerning the use of biological specimens for scientific purposes in compliance with Italian law (art.13 D. Lgs 196/2003). Approval from an ethics committee for virus detection and data publication was not required since data and samples from outpatients with ILI were collected and analyzed anonymously within the national influenza and COVID-19 surveillance.

### 2.2. SARS-CoV-2 RNA Molecular Detection and Quantification

Total RNA was extracted from NPS viavia a commercial kit (QIAamp MinElute Virus Spin Kit, Qiagen, Hilden, Germany) using a semi-automatic extractor (QIAcube, Qiagen, Germany) and tested for the presence of SARS-CoV-2 RNA via a specific one-step rRT-PCR assay according to the Centers for Disease Control and Prevention’ protocol [26]. The result of the rRT-PCR assay was considered positive for SARS-CoV-2 when the Ct was <40 (1.9 log10 copies/mL), whereas we assumed SARS-CoV-2 clearance when Ct ≤ 35 (3.2 log10 copies/mL) [16,27]. To generate the rRT-PCR standard curve for absolute quantification of SARS-CoV-2 RNA, serial 10-fold dilutions of known quantities of an RNA transcript (provided by the European Commission) were performed. The limit of detection (LOD) of the assay for detecting and quantifying SARS-CoV-2 RNA was 2 log10 copies/mL. The same assay was carried out for all samples collected from the two COVID-19 waves.

### 2.3. Statistical Analysis

The analysis described in the following was performed for the 2020 and 2021 winter wave collectives. Categorical variables were reported using counts and percentages; numerical variables were reported using mean and standard deviation or median, 25th, and 75tht centiles, depending on whether the empirical distribution showed a symmetric shape. To evaluate the dynamics of viral load, quantile regression methods were adopted to estimate the median value, the 25th, and the 75th centile of viral load in the days following the date of the index episode. The response variable included in the model was the viral load, expressed in log10 copies/mL. The independent variable included in the model was the individual follow-up time, defined as the number of days that elapsed from the date of the first positive test result (i.e., the start of follow-up). For each subject, the end of the follow-up was set to the date of his/her last positive test result because a negative test result was not observed for everyone. The quantile regression models provide estimates of median and centile viral load distribution on each day after the index episode, considering only the subjects who are still positive on that day. Moreover, since no a priori assumption about the shape of SARS-CoV-2 dynamics was available, we performed a flexible specification of the individual follow-up time using restricted cubic splines [27].

To account for the longitudinal sampling design, the standard errors in each model were calculated by the wild bootstrap technique [28]. This method entailed treating viral load measurements from separate subjects as clustered measurements. Results were reported as estimates of quantiles of viral load distribution at 0, 7, 14, 21, and 28 days from the index episode, with respective 95% CIs. Moreover, histograms were obtained from model results to provide additional insights into the distribution of viral load.

For exploratory purposes, we evaluated putative differences of the median viral load time trend among subgroups related to gender (M/F), age (18–39, 40–64, >64 years), and presence of symptoms (Y/N). In each case, two tests were performed: (1) for the presence of vertical shift; (2) for the difference of shapes. The tests above were performed through the Wald test on regression coefficients, with a significance level set to 5%.

To investigate the time to SARS-CoV-2 clearance, the principal outcome was the time (number of days) elapsed from the date of the first SARS-CoV-2-positive test until the date of SARS-CoV-2 clearance (Ct > 35). Two considerations were crucial for choosing non-parametric methods for interval-censored data [29]:(1)non-infective status was not observed for every subject during the study follow-up. This fact implies using methods of analysis for censored times - that is, survival analysis methods.(2)The exact date of clearance may fall between two consecutive laboratory-testing dates; therefore, it is possible to specify the time interval in which clearance occurred:-for each subject with a laboratory test indicating Ct > 35 at a specific date, the interval consists of the period ending with the date of that test and beginning with the previous testing date;-for each subject with all laboratory tests indicating Ct ≤ 35, we assumed that clearance could occur no longer than 90 days from the index episode. This assumption was justified based on the work of Walker et al. [16] and of Challenger et al. [30].


Results were reported in terms of estimates of the probability that an individual remains with Ct ≤ 35 on each day following the index episode. All the analyses were performed using the software R release 4.1.2 [31] with quantreg package survival packages [32,33] added, and KNIME Analytic Platform version 4.5.0 [34].

## 3. Results

The number of SARS-CoV-2-positive cases (first diagnosis) identified by the regional reference laboratory for COVID-19 was 960 and 330 during the 2020 and 2021 winter epidemic wave (over a total of nearly 9600 and 3500 respiratory samples analyzed), respectively. A total of 262 SARS-CoV-2-positive adults and 784 NPSs were included in our study because, for these individuals, more than one sample had been collected. In total, there were 193 individuals and 593 NPSs in the 2020 wave and 69 individuals and 191 NPSs in the 2021 wave. 

The characteristics of the above two cohorts are reported in Table 1. The viral load was summarized using median and quartiles, as the pertinent data exhibited a pronounced asymmetrical distribution. The individuals in the two periods differed in demographic and clinical characteristics; in the 2020 wave, individuals were basically older and more likely to be symptomatic than in the 2021 wave. Moreover, none were vaccinated against COVID-19 (in Italy, the COVID-19 vaccination campaign started on 27 December 2020), while for the 2021 wave, only vaccinated individuals were included in the study.

### 3.1. Dynamics of SARS-CoV-2 Load

Results concerning the estimated median, 25th, and 75th centiles of SARS-CoV-2 load of positive individuals are reported in Table 2 and Figure 1. Regarding the 2020 winter wave, the median viral load of subjects at the index episode (day 0) was about 6.25 log10 copies/mL (95% CI: 5.50 to 6.70). This corresponds to the median value and respective CI from the biological sample collected at the first laboratory exam for every subject (see statistical analysis paragraph). The curve decreases sharply for about two weeks, reaching the value of 3.40 log10 copies/mL (95% CI: 3.26 to 3.54); this is the estimated median value for subjects still positive after two weeks from the index episode. Furthermore, in the weeks following day 14, the slope of the curve (Figure 1) is much flatter than in the previous period.

Concerning the 25th and 75th centiles, estimates at the index episode are, respectively, 4.10 log10 copies/mL (95% CI: 3.72 to 4.48) and 7.18 log10 copies/mL (95% CI: 6.63 to 7.74). As the time (days) elapsed from the index episode increases, the centiles of viral load decrease similarly with respect to the median viral load curve (Figure 1). Furthermore, the range between these estimates—that is, the interquartile range—decreases, suggesting that the viral load of the subjects in the 2020 wave tends to be more concentrated after the first week from the index episode. This is more evident in Appendix A, which shows the estimated viral load distribution at fixed days from the index episode. 

For the 2021 wave, the median viral load of subjects at the index episode was about 5.42 log10 copies/mL (95% CI: 4.95 to 5.90). Considering that the distinct characteristics of the two populations hinder any comparison between the two waves, in 2021, the median was slightly lower than the estimate for the 2020 wave. The curve has an initial sharp decrease, more accentuated with respect to the 2020 wave, and on day 14, it reaches the value of about 2.93 copies/mL (95% CI: 2.80 to 3.19). The 25th and 75th centiles estimate at the index episode are, respectively, 4.06 log10 copies/mL (95% CI: 3.00 to 5.13) and 7.73 log10 copies/mL (95% CI: 7.22 to 8.23). As previously noted for the 2020 wave, the viral load tends to decrease and to be less concentrated around the median value in the weeks following the index episode and more concentrated in the following periods (Figure 1; Appendix A).

### 3.2. Comparison of Dynamics among Subgroups

The median viral load curves vs. follow-up days are reported in Figure 2 for subgroups concerning age, gender, and presence or absence of symptoms. For what concerns the age classes, the median viral load value was higher at the day of episode index for the youngest (18–39 years) as compared to the older ones (40–64 years and >64 years). A statistically significant difference in viral load shapes was observed in the first wave among age classes (*p* = 0.0302) and symptomatic and asymptomatic subjects (*p* = 0.0187) (Table 3). In particular, the curve for subjects aged >64 years shows a slower decline of median viral load compared to the remainder of the age groups (panel c). In symptomatic individuals, the median viral load at the index episode was higher compared to asymptomatic ones, and the decrease in the viral load in time was higher (panel e). Furthermore, although no statistically significant difference was found for the 2021 wave, the curve pertinent to the younger age class is higher on the date of the index episode than the remainder (panel d).

### 3.3. SARS-CoV-2 Clearance

The probabilities (in percentage) that the Ct value remains under the threshold of 35 in a day following the index episode are reported in Figure 2 and Table 4. For both the epidemic waves, these probabilities show a substantial drop within the first week from the index episode (time = 0–7 days in the figure). On day 7, the probability of being still infective (Ct < 35) was 71.1% (95% CI: 41.6% to 100.0%) in 2020 and 66.2% (95% CI: 30.9 to 100.0%) in the 2021 wave. On day 14, this probability decreased to 50.4% in individuals during the 2020 wave and dropped to 22.3% (95% CI: 6.0% to 82.5%) in 2021 (Table 4). It seems, therefore, that on day 14 in the 2021 wave, the estimated proportion of individuals who can be considered infectious (Ct < 35) is approximately half that of the 2020 wave. This statement, however, should be considered with caution because of the broad confidence intervals, which result in a substantial overlap of estimates between the two periods. The estimated median time for viral clearance in the 2020 wave was about 15 days, while for the 2021 wave it was about 8.5 days.

## 4. Discussion

Understanding the timing of SARS-CoV-2 clearance from the upper respiratory tract and evaluating the putative association between SARS-CoV-2 load trajectories in NPSs and individual/clinical characteristics are pivotal for fine-tuning COVID-19 control strategies, such as the duration of the isolation period and the provision vaccination certificates [35,36]. Two main factors are shaping the COVID-19 pandemic progression, mainly driven by SARS-CoV-2 spread from infected individuals: the emergence of SARS-CoV-2 variants as a result of the massive viral spread; and the coverage of COVID-19 vaccination campaigns.

In this study, we described the SARS-CoV-2 load during the 2020 and the 2021 winter waves, differing with respect to COVID-19 vaccination coverage in the population. These two periods also differed in terms of the circulating SARS-CoV-2 variants and the different rules of quarantine and testing. In Italy, the 2021 wave was characterized by the widespread circulation of the Omicron variant [17], and, in consideration of its epidemiological features, the time for isolation for vaccinated individuals was shortened, and rapid antigen detection tests were used for ending isolation [6].

We described the SARS-CoV-2 loads in the 2020 and 2021 winter waves by estimating the trend of the SARS-CoV-2 load. At the index episode, the median viral load value in the 2021 wave was higher than in the 2020 wave. In contrast, 14 days after the index episode, the median value of viral load in NPS collected from individuals during the 2020 wave was higher than in the 2021 wave. Although any attempt at direct comparison between the two waves would not be properly supported by data—due to the confounding factors related to the different population characteristics, different variants prevalent in the two periods, and different testing options—our results may support data reported by others; individuals infected with the Omicron variant had greater SARS-CoV-2 loads at the index episode than those infected with wild-type virus, thus contributing to increasing the Omicron variant transmissibility [37,38]. In addition, the viral load decline was faster during the 2021 winter than that observed during the 2020 wave; this difference may be attributable to the vaccination, although the intrinsic characteristics of the Omicron variant may also have a biological role [15]. We must consider the in our study settings, the NPSs collected (i.e., symptomatic or asymptomatic individuals) were tested directly via rRT-PCR and were not pre-screened by antigenic test. Thus, this does not affect the inclusion of people with *only* high viral loads who were very symptomatic in our 2021 group. Certainly, in 2021, the use of antigenic tests to end the isolation has reduced the number of cases for whom at least two serial NPSs were collected and tested via rRT-PCR.

We observed no statistically significant differences among COVID-19 vaccinated individuals during the 2021 wave. In contrast, during the 2020 wave, the median viral load value was higher on the day of the index episode for younger adults than the older ones, with symptomatic individuals showing a higher median viral load during the first week of the index episode. However, there are no conclusive results on the different levels of upper respiratory viral load in asymptomatic and symptomatic individuals with SARS-CoV-2 infection [39]. 

The previous comments focus on median viral load; however, inference in quantile regression methods is not restricted to the “typical” (i.e., median) outcome. A further target of this work was to assess the time changes of the determined centiles of viral load to better understand the distribution’s shape at distinct time points. We performed an estimation of the trend in time of the 25th and 75th centiles of viral load. It was impossible to assess the association of these trends with putative risk factors or to extend the results to more extreme outcomes, such as the 5th and 95th centiles of viral load, because these tasks require higher sample sizes. However, results shown here attest to the potential advantages of quantile regression methods over standard ones (e.g., linear regression), focused only on the average individual or outcome.

In our data series, during the 2021 epidemic, the estimated proportion of individuals’ positive results via rRT-PCR with Ct value >35 was approximately half that of the 2020 wave. This suggests that some people may show a slow decrease in viral load after about two weeks, thus keeping a positive status for a relatively long time. In consideration that during the 2021 wave, all individuals included in this study were vaccinated against COVID-19, we can deduce that in our series, the shorter clearance time led to a shorter overall duration of infection among vaccinated individuals, as observed in several cross-sectional studies conducted in other countries [4,12,39,40].

Given all the results from our analyses, this study adds to evidence that individual and viral factors determining SARS-CoV-2 load at the index episode may have a pivotal effect on the viral trajectory during the infection, with faster replication being more complex and offering better control of the infection in terms of reducing SARS-CoV-2 concentration and attaining viral clearance in vaccinated individuals.

The main strengths of this study were the sample size that allowed us to subcategorize the population according to individual and clinical features and the use of statistical methods specific to pursuing the study’s goals in an advanced fashion. The technique allowed for the evaluation of the evolution of SARS-CoV-2 load over time in a broader perspective with respect to the traditional approach of regression to the mean. As is well known, the mean value provides substantial information about the distribution of target variables only in limited cases (i.e., Gaussian distribution). However, in the current research context, the assumption of Gaussian distribution does not “fit”, thus emphasizing the need for more robust analysis methods. Moreover, the method allowed the evaluation of the clearance time using targeted methods that make efficient use of the information obtained as a result of the sampling plan instead of considering the time of laboratory testing, which provides a proxy measure of the true target of interest. As far as we know, the statistical methods adopted in this work have never been used in previous research on this topic. Therefore, a byproduct of our work is our having shown that methods for obtaining more insightful and meaningful results are available to the research community.

### Limitations

The main limitation of this study is that the individuals were predominantly healthy young women and thus were not representative of the general population; to achieve a representative sample, a more comprehensive sampling design was used. The study was retrospective, so a study with a higher level of evidence (prospective) is needed in which swabs should be collected at regular intervals. Moreover, in our study, it was impossible to obtain accurate information about the date of symptom onset for symptomatic subjects. Consequently, it was impossible for symptomatic subjects to study their dynamic starting from symptom onset. Symptoms were tracked during NPSs sampling, but no follow-up data were available. NPSs from individuals were not tested for the presence of infectious viruses by isolation in cells culture. In the symptomatic group, the SARS-CoV-2 load should be elicited by the onset of symptoms. There is no information on the prior immune history of vaccinated and unvaccinated individuals that could influence viral load concentrations and clearance. Concerning the comparison of viral load distribution between subgroups, statistically significant differences emerged only for the first wave. The lack of significant test results for the 2021 wave may be attributed to the expected very low power of statistical tests. More generally, a more accurate analysis for the 2021 wave was unfeasible because of the low number of subjects.

## 5. Conclusions

Our study provides strong insights into SARS-CoV-2 trajectories in upper respiratory tract samples of infected individuals, capturing changes in the epidemiological features of SARS-CoV-2 infection in terms of the time of viral clearance in upper respiratory samples observed in the 2020 and 2021 waves. With the emergence of new variants, understanding SARS-CoV-2 trajectories among people with respect to the presence of symptoms and vaccination status will be pivotal. In public health, studies about the population dynamics of infective agents can assess the duration of an individual’s infectious state. Thus, such studies may provide valuable information for improving the efficiency of lockdown and quarantine protocols. The statistical methods proposed in this work are suitable for the former task and more generally, when a consolidated model for the observed epidemic events is lacking, as is often the case with newly emergent etiological agents.

## Figures and Tables

**Figure 1 viruses-15-01988-f001:**
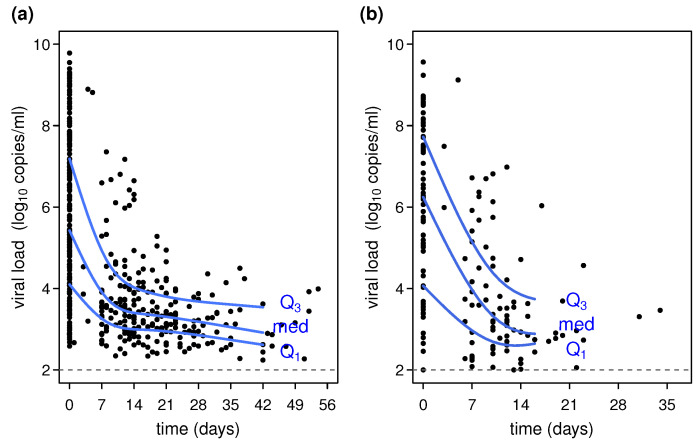
Variations of SARS-CoV-2 load from the day of the episode index. Panel (**a**) 2020 winter wave; panel (**b**) 2021 winter wave. Black dots = recorded viral load from subject examination; blue curves = estimates of median viral load in the first quartile (Q1) and third quartile (Q1). The x-axis represents the time from the episode index, defined as the day of the first positive test result.

**Figure 2 viruses-15-01988-f002:**
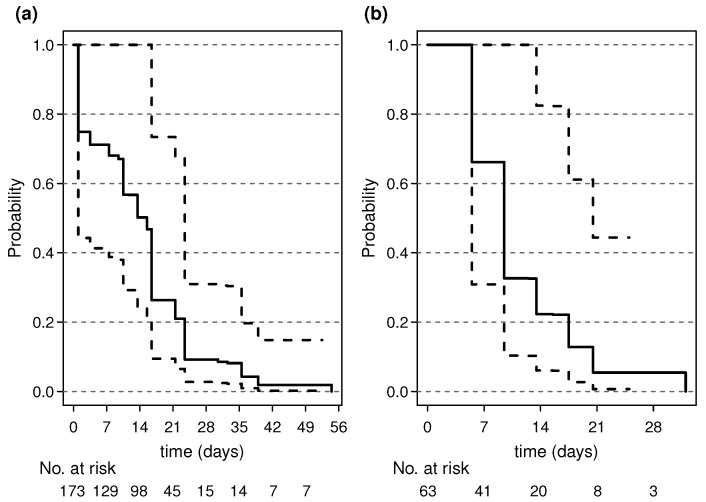
Survival curves showing the estimated probabilities of SARS-CoV-2 clearance over time Panel (**a**) 2020 winter wave; panel (**b**) 2021 winter wave. Solid lines = estimated probabilities; dashed lines = 95% C.I.s.

**Table 1 viruses-15-01988-t001:** Characteristics of the study participants by study group: 2020 winter wave and 2021 winter wave.

	2020 Wave (N = 193)	2021 Wave (N = 69)
**Females, No. (%)**	104 (54.4%)	45 (65.2%)
**Age, mean (sd), (y)**	54.7 (20.1)	45.5 (17.8)
**Age (y)**		
18–39	49 (25.4%)	28 (40.6%)
40–64	86 (44.5%)	32 (46.4%)
>64	58 (30.1%)	9 (13.0%)
**Symptoms, No. (%)**		
symptomatic	142 (73.6%)	22 (31.9%)
asymptomatic	49 (25.4%)	47 (68.1%)
NA	2 (1.0%)	-
**COVID-19 vaccinated: No. (%)**	0 (0%)	69 (100%)
**Number of available serial NPS: No. (%)**		
2	58 (30.1%)	31 (44.9%)
3	87 (45.1%)	26 (37.7%)
4	35 (18.1%)	9 (13.0%)
5	7 (3.6%)	3 (4.3%)
>5	6 (3.1%)	0 (0%)
**Viral load at index episode, median (Q1–Q3) log10 copies/mL**	5.4, 4.1–7.2	6.2, 4.1–7.7

NA = not available; Q1 = first quartile;; Q3 = third quartile.

**Table 2 viruses-15-01988-t002:** Estimates of SARS-CoV-2 viral load from quantile regression models.

Day	Quantile	Estimates	
		2020 Winter Wave Est (95% C.I.)	2021 Winter Wave Est (95% C.I.)
**0**	Q1	4.10 (3.72, 4.48)	4.06 (3.00, 5.13)
	median	5.42 (4.95, 5.90)	6.25 (5.50, 6.70)
	Q3	7.18 (6.63, 7.74)	7.73 (7.22, 8.23)
**7**	Q1	3.26 (3.02, 3.50)	2.97 (2.55, 3.39)
	median	3.91 (3.65, 4.17)	3.97 (3.66, 4.28)
	Q3	4.93 (4.47, 5.39)	5.15 (4.53, 5.77)
**14**	Q1	3.00 (2.87, 3.13)	2.60 (2.19, 3.01)
	median	3.40 (3.26, 3.54)	2.93 (2.80, 3.19)
	Q3	4.03 (3.77, 4.28)	3.85 (3.15, 4.55)
**21**	Q1	2.95 (2.76, 3.13)	NR
	median	3.30 (3.16, 3.45)	NR
	Q3	3.80 (3.53, 4.07)	NR
**28**	Q1	2.85 (2.20, 3.51)	NR
	median	3.18 (3.02, 3.35)	NR
	Q3	3.69 (3.38, 3.98)	NR

Estimated median and quartiles (Q1 = first quartile; Q3 = third quartile) of the distribution and their 95% confidence intervals (C.I.) at the date of the episode index (day = 0) and in following weeks for each of the study groups (2020 wave; 2021 wave). NR = not reported.

**Table 3 viruses-15-01988-t003:** Referring to Appendix A, tests of between-groups differences of median viral load dynamics for each wave.

Tests of Hypothesis			
Subgroups	Test	2020 Winter Wave X^2^, df, *p*	2021 Winter Wave X^2^, df, *p*
**GENDER: F, M**	A	0.20, 1, 0.6557	0.22, 1, 0.6358
	B	0.30, 3, 0.9587	0.24, 3, 0.9716
**AGE: 18–39, 40–64, >64 ****	A	1.92, 2, 0.3817	0,57, 1, 0.4500
	B	13.95, 6, 0.0302 *	5.86, 3, 0.1186
**SYMPTOMATIC: Y, N**	A	2.19, 1, 0.1388	0.26, 1, 0.6126
	B	9.98, 3, 0.0187 *	0.11, 3, 0.9904

Test A: test for vertical shift; Test B: test for shape differences. X^2^ = chi-square test statistic; df = degrees of freedom; *p* = *p*-value; * = *p* < 0.05. ** In the analysis of 2021 wave data, subjects in classes 40–64 and >64 years were grouped into one single class due to an insufficient number of subjects in the >64 years class (see Table 1).

**Table 4 viruses-15-01988-t004:** Estimated probabilities of infectious state (Ct ≤ 35).

Day	Estimates	
	2020 Winter Wave Est (95% CI)	2021 Winter Wave ** Est (95% CI)
**7**	71.1% (41.6%, 100.0%)	66.2% (30.9%, 100.0%)
**14**	50.4% (24.4%, 100.0%)	22.3% (6.0%, 82.5%)
**21**	26.4% (9.5%, 73.0%)	5.4% (0.6%, 44.4%)
**28**	9.2% (2.7%, 31.2%)	NR

Reported in columns 2 and 3 are the estimates of the probability (expressed as a percent) of having upper respiratory tract sample-tested SARS-CoV-2 with a Ct value under 35 at different days from the episode index. NR = not reported. ** The estimates on day 28 were not reported for subjects in the 2021 winter wave.

## Data Availability

Pariani has full access to all the data of this study and takes responsibility for the integrity of the data and the accuracy of data analysis.

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
