# Peer review of "A Flexible Regression Modeling Approach Applied to Observational Laboratory Virological Data Suggests That SARS-CoV-2 Load in Upper Respiratory Tract Samples Changes with COVID-19 Epidemiology"

_viruses, 2023, doi:10.3390/v15101988_

Round 1
Reviewer 1 Report
The submitted manuscript is interesting.
In the introduction, the research problem is well grounded.
The methodology is well described and the methods are correct.
The presentation of the results could be improved.
I recommend that the tabular presentation of the results be more clear and understandable. I mean Table 2 (the hypothesis testing part). It is not very clear what was tested. In my opinion, it is better to present this part in a separate table with more detailed results (number, percentages, etc.).
Author Response
Open Review 1
The submitted manuscript is interesting. In the introduction, the research problem is well grounded. The methodology is well described and the methods are correct. The presentation of the results could be improved.
Q1. I recommend that the tabular presentation of the results be more clear and understandable. I mean Table 2 (the hypothesis testing part). It is not very clear what was tested. In my opinion, it is better to present this part in a separate table with more detailed results (number, percentages, etc.).
R1. Thank you for your comments, we revised the text and table legends. The legend of the table 2 has been integrated as follows:
“Estimates of SARS-CoV-2 viral load from quantile regression models. Estimated median and quartiles (Q1 = first quartile, Q3 = third quartile) of the distribution, and their 95% Confidence Intervals (C.I.), at the date of the episode index (day=0) and following weeks for each of the study groups (2020 wave, 2021 wave).”
The table has been divided in two separate parts, naming Table 3 for the test results. The legend of the Table 3 has been integrated as follows: “Table 3: referring to Supplementary Figure 2, tests of between-groups differences of median viral load dynamics for each wave. Test A: test for vertical shift; Test B: test for shape differences. Est = estimate, C.I. = confidence interval, X2 = chi-square test statistic, df=degrees of freedom, * = p<0.05. NR = Not reported”. Moreover, the comments on the results about Table 3 was linked to the Supplementary Figure 2 and the part was modified as follows:
“3.2 Comparison of dynamics among subgroups“ The median viral load curves vs. follow-up days are reported in Supplementary Figure 2 for subgroups concerning age, gender, and presence or absence of symptoms. For what concerns the age classes, the median viral load value was higher at the day of episode index for the youngest (18-39 years) as compared to the older ones (40-64 years and >64 years). A statistically significant difference in viral load shapes was observed in the first wave among age classes (p=0.0302) and symptomatic and asymptomatic subjects (p=0.0187) (Table 3). In particular, the curve for subjects aged >64 years shows a slower decline of median viral load compared to the remainder age groups (Panel c). In symptomatic individuals, the median viral load at the index episode was higher concerning asymptomatic ones, and the decrease of the viral load in time was higher (Panel e). Furthermore, although no statistically significant difference was found for the 2021 wave, the curve pertinent to the younger age class is higher at the date of the index episode than the remainder (Panel d). “

Reviewer 2 Report
The paper is very interesting and well writen. Methodology, results and conclusions are correcle.
However, I think that an additional paragraph about sensitivity, specificity, positive and negative predicrive value of PCR tests should immprove the quality of the paper. Also, a short comment how validity indices influence serial measurements of viral load.
Also, suggestion for future work, nice alternative to KM curves is recursive partitioning - this method is superb in description of the survival factors. It is available in R as rpart module. Also M5 method for regression may help in some situations - it combines regression and classification tres. It is available in R and Weka data mining system.
OK
Author Response
Open Review 2
The paper is very interesting and well written. Methodology, results and conclusions are correct. However, I think that an additional paragraph about sensitivity, specificity, positive and negative predictive value of PCR tests should improve the quality of the paper. Also, a short comment how validity indices influence serial measurements of viral load.
Q1. Also, suggestion for future work, nice alternative to KM curves is recursive partitioning - this method is superb in description of the survival factors. It is available in R as part module. Also M5 method for regression may help in some situations - it combines regression and classification trees. It is available in R and Weka data mining system.
R1. Thank you for your comments. The partition tree and M5 methods are truly interesting. However, they are suitable when dealing with a large collection of variables and require a sizable case series. These conditions are not met in the current study, where only three variables have been collected.

Reviewer 3 Report
General comments
The manuscript utilizes quantile regression and Kaplan-Meier methods to evaluate the viral load of SARS-CoV-2-positive individuals in two waves of the epidemic in Italy. The authors describe the variants circulating during the two time periods, where the first wave (2020) was before vaccines were available and then the second one when all 69 participants were vaccinated (but not for the variant that was circulating). The reporting of medians and quantiles are appropriate for these non-linear data (without mention of transformations being tried as an alternative) as is quantile regression methods if more non-central results are important for public health measures. The use of robust confidence interval estimation to address the repeated measures obtained on subjects and incorporating the interval censoring as well as right censoring are appropriate for these data. The English grammar could be improved but the manuscript was generally very clearly written.
Abstract:
State the specific methods of analyses here, not that they are innovative. They may be novel in this application area but are not considered innovative methods as quantile regression is being used frequently now.
The cut off for an rRT-PCR positive assay result is stated as (Ct<35) here but in the methods it is (Ct<40).
The conclusion does not indicate the real benefits of applying this method over other methods that could be used early on in a new outbreak. This is related to my comments in the discussion and results sections where the benefits of quantile regression were not articulated.
Keywords: use quantile regression model not regression modelling approach, add in Kaplan-Meier curve
Introduction:
The previous research cited covers different variants of the SARS-CoV-2 virus, as well as when vaccinations were/were not available and had different uptake in the populations. Citing some specific previous studies during the same time period as what this study reports on would improve the comparisons between this study and other ones. Noting the differences that could drive results would be helpful to the reader to put this study into context.
The statement on lines 83-4 does not acknowledge there are other differences between people who are vaccinated or not, namely the predominant variants in circulation during the time period in question. It is an important issue, but this study cannot fully answer this question as vaccination status is confounded with the variants in circulation between the two study cohorts. Revise this sentence to make it clearer that this study cannot fully answer this question. Line 105 does clarify this but the sentence in the introduction suggests this study can address this question.
The statements on lines 87-90 are appropriate for populations with access to vaccinations or when the previous vaccination still confers protection. It is not a universal statement for all populations.
Materials and Methods
The cut off for an rRT-PCR positive assay result was for Ct <40 in the methods, but stated as Ct < 35 in the abstract. Please confirm which cutoff was used and revise the corresponding text to make it consistent.
Two different tests were used in each wave to determine if an individual was positive for COVID-19 was not described. Would these two different methods impact the results? Some comments are needed here to ensure the same outcomes are being compared as the timing to obtain a test result would be different (affects lead infection time in the data).
Results:
Why was only about 1/5 of the available participants enrolled in this study from the available samples? Please provide details on the substantial missing data as the reasons could determine if they are missing at random or not at random.
The introduction and methods state that the two study periods cannot really be compared, yet they are in the results without noting the substantial differences in the populations.
The two study populations are very different in their characteristics too, especially whether they were symptomatic or not. Study populations includes both diagnostic and routine surveillance patients who would have very different probabilities of being infected. This also makes the comparisons between years also challenging, and should be noted as another source of confounding when discussing the results.
On line 156 the mean and standard deviation would be reported if the empirical distribution showed symmetry. Were these values ever reported? I did not see any results with means or sds so can just state the median and quantiles are reported as the data did not appear to be normally distributed.
On lines 178-80, two different tests were described in the subgroup analyses. Table 2 is clear about which test was statistically significant but the text in Section 3.2 was not that clear if the higher viral load for the 2020 wave only affected the shape, not the shift. This section could be rewritten to make the results clearer, with the caveat that the subgroups are really small. This is especially relevant when utilizing a semi-parametric method like quantile regression that needs a lot of data for estimating the model parameters. Thus, non-significant results are not necessarily surprising and should be reported as such.
In Section 3.3 SARS-CoV-2 clearance the cutoff for being infective is now Ct < 35. Was more than one cutoff used? This is unclear due to the inconsistencies reported in different parts of the manuscript.
Lines 223-4 – this sentence belongs in the discussion section.
Line 246: rewite as “The probabilities (in percentages) that … “ as probabilities are typically reported as a value in [0,1].
Table 3 and Section 3.3 – the confidence intervals are extremely wide with substantial overlap between the two years. This point needs to be included, not just reporting the CIs, as the statement on lines 252-4 suggesting the proportion of people still considered infectious is misleading. The title for Table 3 does not seem to match what is stated in the legend (testing positive versus clearing the virus). Figure 2 can include the right censoring marks as that would also be helpful in seeing the amount of data being used to estimate these curves. It is typical to report the median survival time too from Kaplan-Meier curves and could be included here. This comparison could be moved to the supplementary materials to better reflect the confidence you have in the results.
Discussion
The results and discussion do not make use of the quantile regression model adopted in this study. Is it only the median or average time in a population that matters? What about reporting the third quartile, when ¾ of the people would have a sufficiently lower viral load as that might be relevant to public health policies? The value of utilizing quantile regression is not really exploited. The value of this approach compared to previous studies are not clearly articulated, so the value of this work in the context of so many other studies on viral load is not that clear.
Lines 315-321 – the subgroup analyses were described as exploratory and the really small subgroups sizes for 2021 almost guarantee nothing would be statistically significant in a quantile regression model. This caveat needs to be included here as the reasons for the non-significant results are likely due to low statistical power. Line 336 is really only for 2020, not 2021 and needs to be clarified as such.
Lines 340-4: did you try a transformation of your responses to see if a Gaussian model assumption could hold? If a transformation could be found, linear mixed models would be more efficient than a quantile regression model without such an assumption.
The reference on line 331 is incorrect. The study you are comparing too needs to cover the same time periods and prevalent variants to make the comparison meaningful.
Line 353: this statement contradicts what is presented in Table 1, as the majority of participants were females.
Conclusion: the statement on lines 372-3 could be true if many participants who were positive were obtained in short period of time. The premise that this modelling approach will help will require substantial data collection early in the infection wave so might not be that practical to implement. Furthermore, the value of knowing when 25% or 75% of the people could have a sufficiently low viral load is not described at all but could be relevant to public health officials.
Some minor edits are required, such as on line 60 (why is there a ']'? Line 116 should be tested not resulted? Nonparametric is more commonly used than non-parametric. Line 273 it is Table 3 not Tab. 3. Some third person voice used, for example on line 349.
Author Response
Open Review 3
The manuscript utilizes quantile regression and Kaplan-Meier methods to evaluate the viral load of SARS-CoV-2-positive individuals in two waves of the epidemic in Italy. The authors describe the variants circulating during the two time periods, where the first wave (2020) was before vaccines were available and then the second one when all 69 participants were vaccinated (but not for the variant that was circulating). The reporting of medians and quantiles are appropriate for these non-linear data (without mention of transformations being tried as an alternative) as is quantile regression methods if more non-central results are important for public health measures. The use of robust confidence interval estimation to address the repeated measures obtained on subjects and incorporating the interval censoring as well as right censoring are appropriate for these data. The English grammar could be improved but the manuscript was generally very clearly written.
Q1. Abstract: State the specific methods of analyses here, not that they are innovative. They may be novel in this application area but are not considered innovative methods as quantile regression is being used frequently now.
R1. Thank you for your suggestion. We modified the abstract as follows: “1) Background. Exploring the evolution over time of SARS-CoV-2 load and clearance from the upper respiratory tract samples is of foremost importance to improve Covid-19 control. Data were collected retrospectively from a laboratory dataset on SARS-CoV-2 load quantified in leftover nasal-pharyngeal swabs (NPSs) collected from symptomatic/asymptomatic individuals who resulted positive to SARS-CoV-2 RNA detection in the framework of testing activities for diagnostic/screening purpose during the 2020 and 2021 winter epidemic waves .2). Methods. statistical approach (quantile regression and survival models for interval censored data), novel for this kind of data, was applied. We included in the analysis SARS-CoV-2-positive adults >18 years old for whom at least 2 serial NPSs were collected. 262 SARS-CoV-2-positive individuals and 784 NPSs were included: 193 (593 NPSs) during the 2020 winter wave (before COVID-19 vaccine introduction) and 69 (191 NPSs) during the 2021 winter wave (all COVID-19 vaccinated). We estimated the trend of the median value, 25-th and 75-th centiles of the viral load, from the index episode (i.e. first SARS-CoV-2-positive test) until the 6-th week (2020 wave) and the 3-rd week (2021 wave). Interval censoring methods were used to evaluate the time to SARS-CoV-2 clearance (defined as Ct<35) […]”
Q2. The cut off for an rRT-PCR positive assay result is stated as (Ct<35) here but in the methods it is (Ct<40).
R2. The cut off of rRT-PCR positive assay result is Ct<40 whereas the cut off for considering an individual infectious is Ct<35 (please refer to reference n. 25). We have clarified this issue in the text.
Q3. The conclusion does not indicate the real benefits of applying this method over other methods that could be used early on in a new outbreak. This is related to my comments in the discussion and results sections where the benefits of quantile regression were not articulated.
R3. Thank you for your suggestion. We have added, in the conclusion, the following sentences: “With the emergence of new variants, understanding SARS-CoV-2 trajectories among people concerning the presence of symptoms and vaccination status will be pivotal. In public health, studies about the population dynamics of infective agents can assess the duration of an individual’s infectious state. Thus, such studies may provide a piece of valuable information for improving the efficiency of lockdown and quarantine protocols. The statistical methods proposed in this work are suitable for the earlier task, and, more generally, when a consolidated model for the observed epidemic events is lacking, as is often the case with newly emergent etiological agents.”
Q4. Keywords: use quantile regression model not regression modelling approach, add in Kaplan-Meier curve
R4. Thank you for your suggestion. We have added“Kaplan-Meier curve” as a keyword.
Q5. The previous research cited covers different variants of the SARS-CoV-2 virus, as well as when vaccinations were/were not available and had different uptake in the populations. Citing some specific previous studies during the same time period as what this study reports on would improve the comparisons between this study and other ones. Noting the differences that could drive results would be helpful to the reader to put this study into context.
R5. Thanks for your comments, we have added new references (i.e references n. 23-24).
Q6. The statement on lines 83-4 does not acknowledge there are other differences between people who are vaccinated or not, namely the predominant variants in circulation during the time period in question. It is an important issue, but this study cannot fully answer this question as vaccination status is confounded with the variants in circulation between the two study cohorts. Revise this sentence to make it clearer that this study cannot fully answer this question. Line 105 does clarify this but the sentence in the introduction suggests this study can address this question.
R6. Thanks for your suggestion. We have revised the sentence as follow: “In population with high Covid-19 vaccination coverage approaching to return to normality and in consideration of the predominant circulating variants of SARS-CoV-2, it is important to evaluate whether viral shedding duration may have different trajectories in unvaccinated and vaccinated individuals, to apply appropriate control measures. This appears even more relevant considering that in population with access to vaccination, vaccines should reduce SARS-CoV-2 transmission by affecting the viral load titre and viral clearance in the upper respiratory samples, as reported by several studies although effects of vaccines should be confounded by the circulating predominant SARS-CoV-2 variant [13, 19-22].This appears even more relevant considering that vaccination should reduce the SARS-CoV-2 transmission by affecting the viral load titre and viral clearance in the upper respiratory samples, as reported by several studies [13, 19-22]. Therefore, understanding the timing of SARS-CoV-2 clearance in the upper respiratory tract samples in a highly vaccinated population is pivotal to inform public health stakeholders on effective mitigation strategies, such as defining isolation and quarantine time and to support the implementation of restrictive measures.
Q7. The statements on lines 87-90 are appropriate for populations with access to vaccinations or when the previous vaccination still confers protection. It is not a universal statement for all populations.
R7. We agree with this consideration. We have changed the sentence as follow: “In population with high Covid-19 vaccination coverage approaching to return to normality, it is important to evaluate whether viral shedding duration may have different trajectories in unvaccinated and vaccinated individuals, to apply appropriate control measures. This appears even more relevant considering that in population with access to vaccination, it has been demonstrated that vaccines reduce SARS-CoV-2 transmission by affecting the viral load titre and viral clearance in the upper respiratory samples, as reported by several studies [13, 19-22].
Q8. The cut off for an rRT-PCR positive assay result was for Ct <40 in the methods, but stated as Ct < 35 in the abstract. Please confirm which cutoff was used and revise the corresponding text to make it consistent.
R8. The cut off of rRT-PCR positive assay result is Ct<40 whereas the cut off for considering an individual infectious is Ct<35 (please refer to reference n. 25). We have clarified this issue in the text.
Q9. Two different tests were used in each wave to determine if an individual was positive for COVID-19 was not described. Would these two different methods impact the results? Some comments are needed here to ensure the same outcomes are being compared as the timing to obtain a test result would be different (affects lead infection time in the data).
R9. The same assay was carried out for all samples collected from the two COVID-19 waves. We think it is a strength of the work. We have detailed it in the material section.
Q10. Why was only about 1/5 of the available participants enrolled in this study from the available samples? Please provide details on the substantial missing data as the reasons could determine if they are missing at random or not at random.
R10. We have specified it as follow: “because for these individuals more than one sample was collected”
Q11. The introduction and methods state that the two study periods cannot really be compared, yet they are in the results without noting the substantial differences in the populations.
R11. We have added this in the part 3.1 of the results: “Considering that the distinct characteristics of the two reference populations hinder any comparison between the two waves, in 2021, the median was slightly lower than that estimated for the 2020 wave.”
Q12. The two study populations are very different in their characteristics too, especially whether they were symptomatic or not. Study populations includes both diagnostic and routine surveillance patients who would have very different probabilities of being infected. This also makes the comparisons between years also challenging, and should be noted as another source of confounding when discussing the results.
R12. Thanks for this comment. We have added the following sentence in the discussion section:” Although any attempt of direct comparison between the two waves would not be properly supported by data - due to the confounding factors related to the different population characteristics, different prevalent variants in the two periods, and different testing options - results may support data reported by others…”
Q13. On line 156 the mean and standard deviation would be reported if the empirical distribution showed symmetry. Were these values ever reported? I did not see any results with means or sds so can just state the median and quantiles are reported as the data did not appear to be normally distributed.
R13. At the beginning of paragraph 3 of the results , we have added: “The characteristics of the above two cohorts are reported in Table 1. The viral load was summarized using median and quartiles, as the pertinent data exhibited a pronounced asymmetrical distribution. “
Q14. On lines 178-80, two different tests were described in the subgroup analyses. Table 2 is clear about which test was statistically significant but the text in Section 3.2 was not that clear if the higher viral load for the 2020 wave only affected the shape, not the shift. This section could be rewritten to make the results clearer, with the caveat that the subgroups are really small. This is especially relevant when utilizing a semi-parametric method like quantile regression that needs a lot of data for estimating the model parameters. Thus, non-significant results are not necessarily surprising and should be reported as such.
R14. We have modified the description of the results, reporting that only the difference in shape was significant for the 2020 wave. The statistical test was significant, because of the putative greater power with respect to the 2021 wave (lower number of subjects). Nevertheless, results related to Supplementary Figure 2, are represented and commented independently from statistically significance. The draft was modified, also accounting for the comments made by referee 1:
The legend of the table 2 has been integrated as follows:
“Estimates of SARS-CoV-2 viral load from quantile regression models. Estimated median and quartiles (Q1 = first quartile, Q3 = third quartile) of the distribution, and their 95% Confidence Intervals (C.I.), at the date of the episode index (day=0) and following weeks for each of the study groups (2020 wave, 2021 wave).”
The table has been divided in two separate parts, naming Table 3 for the test results. The legend of the Table 3 has been integrated as follows:
“Table 3: referring to Supplementary Figure 2, tests of between-groups differences of median viral load dynamics for each wave. Test A: test for vertical shift; Test B: test for shape differences. Est = estimate, C.I. = confidence interval, X2 = chi-square test statistic, df=degrees of freedom, * = p<0.05. NR = Not reported”.
Moreover the comments on the results about Table 3 was linked to the Supplementary Figure 2 and the part was modified as follows: “3.2 Comparison of dynamics among subgroups” “The median viral load curves vs. follow-up days are reported in Supplementary Figure 2 for subgroups concerning age, gender, and presence or absence of symptoms. For what concerns the age classes, the median viral load value was higher at the day of episode index for the youngest (18-39 years) as compared to the older ones (40-64 years and >64 years). A statistically significant difference in viral load shapes was observed in the first wave among age classes (p=0.0302) and symptomatic and asymptomatic subjects (p=0.0187) (Table 3). In particular, the curve for subjects aged >64 years shows a slower decline of median viral load compared to the remainder age groups (Panel c). In symptomatic individuals, the median viral load at the index episode was higher concerning asymptomatic ones, and the decrease of the viral load in time was higher (Panel e). Furthermore, although no statistically significant difference was found for the 2021 wave, the curve pertinent to the younger age class is higher at the date of the index episode than the remainder (Panel d).
“Moreover, in section 4.1 (limitations of the study), we added the following consideration:
“Concerning the comparison of viral load distribution between subgroups, statistically significant differences emerged only for the first wave. The lack of significant test results for the 2021 wave may be attributed to the expected very low power of statistical tests. More generally, a more accurate analysis for the 2021 wave was unfeasible because of the low number of subjects”
Q15. In Section 3.3 SARS-CoV-2 clearance the cut off for being infective is now Ct < 35. Was more than one cut off used? This is unclear due to the inconsistencies reported in different parts of the manuscript.
R15. The result of the rRT-PCR assay was considered positive for SARS-CoV-2 when the Ct was <40 whereas we assumed SARS-CoV-2 clearance when Ct ≤ 35 [16, 27].
Q16. Lines 223-4 – this sentence belongs in the discussion section.
R16. We have revised the sentence accordingly.
Q17. Line 246: re write as “The probabilities (in percentages) that … “ as probabilities are typically reported as a value in [0,1].
R17. We have revised the sentence accordingly.
Q18. Table 3 and Section 3.3 – the confidence intervals are extremely wide with substantial overlap between the two years. This point needs to be included, not just reporting the CIs, as the statement on lines 252-4 suggesting the proportion of people still considered infectious is misleading.
R18. Thanks for this suggestion. We have modified the test in the result part, section 3.3 as follows: “It seems, therefore, that at day 14 in the 2021 wave, the estimated proportion of individuals who can be considered infectious (Ct<35) is approximately half as compared to the 2020 wave. This statement, however, should be considered with caution because of the broad confidence intervals, which result in substantial overlap of estimates between the two periods.”
Q19. The title for Table 3 does not seem to match what is stated in the legend (testing positive versus clearing the virus).
R19. We have modified the title and the legend of the table as follows: “Tab. 4. Estimated probabilities of infectious state (Ct≤35)” “Reported in columns 2 and 3 there are the estimates of the probability (expressed as a percent) of having an upper respiratory tract sample tested SARS-CoV-2 with a Ct value under 35 on different days from the episode index”.
Q20. Figure 2 can include the right censoring marks as that would also be helpful in seeing the amount of data being used to estimate these curves. It is typical to report the median survival time too from Kaplan-Meier curves and could be included here. This comparison could be moved to the supplementary materials to better reflect the confidence you have in the results.
R20. Instead of using the pair (t, d), where t represents the observed individual time and d is the censoring indicator, the standard notation for interval-censored data consists of a time interval: (tLOWER, tUPPER), indicating that the exact event time is known to have occurred within this range. For our analysis, this representation had to be adopted for each subject in the study; how time intervals details were specified is illustrated in detail in the methods section. For the reasons above, censoring times are not relevant in interval-censored methods. According to our knowledge, right censoring times are not reported on the Kaplan-Meier curve plots. Estimated median times:
2020 wave: about 15 days
2021 wave: about 8.5 days
In the section 3.3 the following sentence has been added: “The estimated median time for viral clearance in 2020 wave were about 15 days, and for 2021 wave were about 8.5 days”
Q21. The results and discussion do not make use of the quantile regression model adopted in this study. Is it only the median or average time in a population that matters? What about reporting the third quartile, when ¾ of the people would have a sufficiently lower viral load as that might be relevant to public health policies? The value of utilizing quantile regression is not really exploited. The value of this approach compared to previous studies are not clearly articulated, so the value of this work in the context of so many other studies on viral load is not that clear.
R21. The reported results are not referred to the quantile of the distribution of viral load, in which the median is 50% of people and so on. Still, it is referred to as the estimates of quantiles and at its 95% confidence intervals. The assessment of lower and upper limits, such as the 2.5% and 97.5% quantiles, is indeed very appealing. However, it is impossible to achieve this task due to the low number of participants who showed a sufficiently high viral load after weeks from the episode index. Empirically, we decided to estimate 25% and 75% quantiles because these indices are typically used to describe the dispersion of data around the median. Moreover, this approach allows us to obtain straightforward estimates and confidence intervals of the differences in the quantiles among subgroups.
For what concerns the value of the proposed approach, we added in the conclusion section the following consideration: “In public health, studies about the population dynamics of infectious agents can assess the duration of an individual’s infectious state. Thus, such studies may provide valuable information for improving the efficiency of lockdown and quarantine protocols. The statistical methods proposed in this work are suitable for the earlier task, and, more generally, when a consolidated model for the observed epidemic events is lacking, as is often the case with newly emergent etiological agents”
Q22. Lines 315-321 – the subgroup analyses were described as exploratory and the really small subgroups sizes for 2021 almost guarantee nothing would be statistically significant in a quantile regression model. This caveat needs to be included here as the reasons for the non-significant results are likely due to low statistical power. Line 336 is really only for 2020, not 2021 and needs to be clarified as such.
R22. In section 4.1, we have added the following sentence: “Concerning the comparison of viral load distribution between subgroups, statistically significant differences emerged only for the first wave. The lack of significant test results for the 2021 wave may be attributed to the expected very low power of statistical tests. More generally, a more accurate analysis for the 2021 wave was unfeasible because of the low number of subjects”
Q23. Lines 340-4: did you try a transformation of your responses to see if a Gaussian model assumption could hold? If a transformation could be found, linear mixed models would be more efficient than a quantile regression model without such an assumption.
R23. We were not interested in using a transformation of the response variable. We have decided to use the quantile model because of the straightforward interpretation of model results, avoiding retro-transformation of estimates and 95% Confidence Intervals. We accounted for the correlation among viral load repeated measurements using the “wild bootstrap” technique. For better clarification of the use of this technique with longitudinal data, we revised the following sentence in the statistical methods section: “To account for the longitudinal sampling design, the standard errors in each model were calculated by the wild bootstrap technique [25]. This entailed treating measurements of viral load from separate subjects as clustered measurements. “
Q24. The reference on line 331 is incorrect. The study you are comparing too needs to cover the same time periods and prevalent variants to make the comparison meaningful.
R24. Thanks for this comment, we have changed the reference.
Q25. Line 353: this statement contradicts what is presented in Table 1, as the majority of participants were females.
R25. We have amended it.
Q26. The statement on lines 372-3 could be true if many participants who were positive were obtained in short period of time. The premise that this modelling approach will help will require substantial data collection early in the infection wave so might not be that practical to implement. Furthermore, the value of knowing when 25% or 75% of the people could have a sufficiently low viral load is not described at all but could be relevant to public health officials.

Round 2
Reviewer 3 Report
Most of my comments have been addressed by the authors in their responses as well as in the revised manuscript. However, a few comments still remain and need to be addressed:
1. the word 'quantile' does not appear in the title or in the keywords (my previous suggestion) so that will impact the results for anyone searching using this term. I still suggest it adds important value to your study to include it with the phrase 'regression modelling approach.' This would support your desire to introduce this appropriate method to the infectious disease community.
2. My previous question (Q9) about the tests that were used in each wave to determine if an individual was positive for COVID-19 referred to the tests used for surveillance and diagnosis not confirmation of positivity status. Using the same confirmation rRT-PCR assay is indeed a strength of this paper. Please add in a comment about whether the different surveillance tests affected your results as these tests have different levels of sensitivity and specificity. It does affect who then gets tested to confirm their infection status, which could impact the populations you studied for each year. The antigenic tests have lower sensitivity so only people with high viral loads and were very symptomatic were likely included in your 2021 cohort. See https://www.cdc.gov/coronavirus/2019-ncov/lab/resources/antigen-tests-guidelines.html
3. My previous question (Q21) refers to the lack of discussion about quantiles other than the median; it was not really understood or addressed. Quantile regression results for the 25th and 75th percentiles (Q1 and Q3 from Figure 2 and Table 2) can tell public health leaders a lot more than just reporting the median values with confidence intervals. The median or average person is important but the subgroup of people in the first and third quantiles are likely of interest too. For example, the 2021 data in Table 2 really shows how much higher the viral load is in the Q3 subgroup. This would add to your conclusions that only focus on average people as often policies are based on when most people would no longer be positive not just half of them. It also strengthens your conclusions for utilizing quantile regression for this type of modelling over other methods that only consider mean values and linear functions.
4. As I was rereading the revised manuscript, I realized I did not see what the viral load in log base 10 copies/ml would correspond to the Ct values of 35 and 40. Could this information please be added? It would make understanding the supplementary figure 1 easier to understand. I expected Ct value of 35 would be about 2.53 on the log 10 scale but would rather see your calculated values to ensure proper understanding of the results.
Grammar is generally fine but there are still a number of grammatical errors that need to be corrected to polish this paper.
Author Response
Ms. Ref. Title: A flexible regression modelling approach applied to observational laboratory virological data suggests that SARS-CoV-2 load in upper respiratory tract samples changes with Covid-19 epidemiology
Journal: Viruses
Dear Editor,
we would like to thank you and the reviewers for your helpful comments. We have addressed all the points raised in a point-by-point response. We believe that the manuscript has been improved by the revision and we hope it now meets with your approval to be published in Viruses Journal.
Kindest regards,
Elena Pariani
Elena Pariani, BS, PhD
Department of Biomedical Sciences for Health, University of Milan, Milan
Via Carlo Pascal, 36 – 20133, Milano – Italy
Phone: +39 0250315132
Fax: +39 0250315121
e-mail: elena.pariani@unimi.it
Open Review 3
Most of my comments have been addressed by the authors in their responses as well as in the revised manuscript. However, a few comments still remain and need to be addressed.
Q1. The word 'quantile' does not appear in the title or in the keywords (my previous suggestion) so that will impact the results for anyone searching using this term. I still suggest it adds important value to your study to include it with the phrase 'regression modelling approach.' This would support your desire to introduce this appropriate method to the infectious disease community
R1. We apologise we have missed it in our previous revision. The terms ‘quantile regression’ and ‘regression modelling approach’ have been added to the keywords
Q2. My previous question (Q9) about the tests that were used in each wave to determine if an individual was positive for COVID-19 referred to the tests used for surveillance and diagnosis not confirmation of positivity status. Using the same confirmation rRT-PCR assay is indeed a strength of this paper. Please add in a comment about whether the different surveillance tests affected your results as these tests have different levels of sensitivity and specificity. It does affect who then gets tested to confirm their infection status, which could impact the populations you studied for each year. The antigenic tests have lower sensitivity so only people with high viral loads and were very symptomatic were likely included in your 2021 cohort. See https://www.cdc.gov/coronavirus/2019-ncov/lab/resources/antigen-tests-guidelines.html
R2. In our series, the same rRT-PCR assay was applied for detecting SARS-CoV-2 genome in all investigated samples collected during the two COVID-19 waves for surveillance and diagnosis. Molecular testing was conducted for diagnostic purposes in symptomatic individuals (i.e., patients with respiratory symptoms who seek medical advice or are hospitalized with Covid-19) and for case-finding/screening purposes in asymptomatic individuals (i.e., healthcare personnel and nursing home residents tested every two weeks). As reported in materials and methods section (lines 132----) there were differences in SARS-CoV-2 testing options during the two study periods: i) in 2020, only molecular tests were used in diagnostic and surveillance settings, whereas antigenic tests were used as an alternative in 2021; ii) testing for ending isolation was carried out mainly with antigenic test in 2021.
In the settings where samples were collected (i.e. symptomatic or asymptomatic individuals), all specimens were tested directly by rRT-PCR and were not pre-screened by antigenic test. Thus, we believe that this does not affect the inclusion of only people with high viral loads and very symptomatic in our 2021 group). For sure, in 2021 the use of antigenic tests to end the isolation has reduced the number of cases for whom at least 2 serial NPSs were collected and tested by rRT-PCR.
Q3. My previous question (Q21) refers to the lack of discussion about quantiles other than the median; it was not really understood or addressed. Quantile regression results for the 25th and 75th percentiles (Q1 and Q3 from Figure 2 and Table 2) can tell public health leaders a lot more than just reporting the median values with confidence intervals. The median or average person is important but the subgroup of people in the first and third quantiles are likely of interest too. For example, the 2021 data in Table 2 really shows how much higher the viral load is in the Q3 subgroup. This would add to your conclusions that only focus on average people as often policies are based on when most people would no longer be positive not just half of them. It also strengthens your conclusions for utilizing quantile regression for this type of modelling over other methods that only consider mean values and linear functions.
R3. Thank you for your suggestion. To address this topic, we have re-written some comments in the results section (page 6) as follows: “Concerning the 25-th and 75-th centiles, estimates at the index episode are, respectively, 4.10 log10 copies/ml (95% CI: 3.72 to 4.48) and 7.18 log10 copies/ml (95% CI: 6.63 to 7.74). As the time (days) elapsed from the index episode increases, the centiles of viral load decrease similarly with respect to the median viral load curve (Figure 1). Furthermore, the range between these estimates – that is, the interquartile range – decreases, suggesting that the viral load of the subjects in the 2020 wave tends to be more concentrated after the first week from the index episode. This is more evident in Supplementary Figure 1A, which shows the estimated viral load distribution at fixed days from the index episode.
The 25-th and 75-th centiles estimates at the index episode are, respectively, 4.06 log10 copies/ml (95% CI: 3.00 to 5.13) and 7.73 log10 copies/ml (95% CI: 7.22 to 8.23). As previously noted for the 2020 wave, the viral load tends to decrease and to be less concentrated around the median value in the weeks following the index episode and more concentrated in the following periods (Figure 1, Supplementary Figure 1B)”
The following sentences were added in the discussion (page 12), for strengthening the conclusions about the utility of quantile regression methods: “The previous comments focus on median viral load; however, inference in quantile regression methods is not restricted to the "typical" (i.e., median) outcome. A further target of this work was to assess the time changes of determined centiles of viral load better to understand the distribution's shape at distinct time points. We performed an estimation of the trend in time of 25th and 75th centiles of viral load. It was impossible to assess the association of these trends with putative risk factors or to extend the results to more extreme outcomes, such as 5-th and 95-th percentiles of viral load, because these tasks require higher sample sizes. However, results shown here attest to the potential advantages of quantile regression methods to standard ones (e.g., linear regression), focused only on the average individual or outcome.”
Q4. As I was rereading the revised manuscript, I realized I did not see what the viral load in log base 10 copies/ml would correspond to the Ct values of 35 and 40. Could this information please be added? It would make understanding the supplementary figure 1 easier to understand. I expected Ct value of 35 would be about 2.53 on the log 10 scale but would rather see your calculated values to ensure proper understanding of the results
R4. Thank you for your suggestion. The Ct value of 35 corresponds to 3.2 log10 copies/ml and Ct of 40 corresponds to 1.9 log10 copies/ml. We have added this information in the text.
Q5. Grammar is generally fine but there are still a number of grammatical errors that need to be corrected to polish this paper
R5. The paper has been proofread to amend typos and grammatical errors
